# Spiritual Diversity in Personal Recovery from Mental Health Challenges: A Qualitative Study from Chinese-Australian Service Users’ Perspectives

**DOI:** 10.3390/ijerph20032210

**Published:** 2023-01-26

**Authors:** Ling He, Melissa Petrakis

**Affiliations:** 1Psychosocial Support Service, Wellways Australia, Melbourne, VIC 3132, Australia; 2Department of Social Work, Monash University, Melbourne, VIC 3145, Australia; 3Mental Health Service, St. Vincent’s Hospital Melbourne, Melbourne, VIC 3065, Australia

**Keywords:** personal recovery, spiritual diversity, service users, mental health, Chinese

## Abstract

Spiritual diversity and the positive role of spirituality in personal recovery have received growing attention in mental health literature. However, the spiritual experiences and views of service users from cultural communities, particularly the Chinese community, are understudied in Australia. This research explores Chinese service users’ spiritual identities and their views and perspectives on the roles of spirituality in their mental health recovery and attempts to provide inspiration for practitioners to engage with service users’ spirituality in non-clinical mental health practice. A qualitative exploratory approach guided this study. Semi-structured interviews were employed with four Chinese participants with spiritual identities, who were recruited through the community Psychosocial Support Service in Victoria. The template analysis method supported the data analysis. The results indicate that spirituality has a positive impact on the mental health recovery of participants, primarily through coping, self-regulatory, and social support mechanisms. The findings also present that Chinese service users’ understanding and approaches to spirituality are shaped by both original and Australian Cultures. These findings suggest that practitioners should provide a creative understanding and cultural awareness when discussing with service users their spiritual identities, perspectives, and spirituality in the wider context. The research fills a gap in the spiritual views and perspectives of service users accessing a non-clinical mental health service from the Chinese community.

## 1. Introduction

Increasing evidence indicates that spirituality and religion are significant in promoting the mental health recovery of people experiencing mental health challenges by enabling coping, making meaning, and providing a sense of purpose, social support, and other mechanisms [1,2,3]. However, spirituality and religion may present challenges to mental health, for example, spiritual stigma, discrimination, or/and conflicting views from within and beyond religious communities that hinder recovery or may exacerbate symptoms [3,4,5]. Pargament and Lomax [6] argued that, given the potentially constructive or/and destructive impacts on mental health, it is essential to explore the broader dimensions of spirituality and the associated themes for people from diverse cultures and religions.

The positive contributions of spirituality and religion to mental health, through various mechanisms, have been widely evidenced [2,3,4]. However, the understanding and perception of spirituality and the mechanism through which it acts for mental recovery may vary across cultural groups. For instance, a study of Latino immigrants in the United States revealed that the religious identity and the social support that derives from their connection to the religious community are some of the major resources for the US Latino populations to encounter uncertainty or mental challenges [7]. Additionally, Baijnath’s [8] qualitative study on two separate cultural groups found that religious identity and regular religious practices played a significant role in the mental health recovery of Australian- Indians experiencing mental challenges by enabling inner peace and allowing the meaning of life to be interpreted, while Australian Anglos were less likely to associate mental illness with their spiritual identity. Despite the differences in the mechanisms of action, people in faith communities appear to perceive their religious identity as a contributor to mental health recovery to a deeper degree.

It has been suggested that a measure of spiritual identity—separate from a religious focus—is needed to observe and capture the range of practices, attitudes, and emotional reactions in people’s different spiritual identities [9]. However, the individuality, inconsistency, and increasing complexity of spiritual identities cause difficulties in describing or defining spirituality, which may also account for the challenge of conducting statistics on non-religious but self-identified spiritual populations [10,11]. For example, in Hong Kong, people with mental conditions incorporate their cultural identity or traditional Eastern values—spiritual beliefs that lead them to find peace and inner harmony to cope with mental challenges—based on whether they have a religious background [12]. Thus, rather than conceptualizing and defining spirituality, the current research attempts to interpret ‘spirituality’ abstractly as the unique inner experiences people describe that involve meaning, hope, well-being, and the way that personal values and cultural identity influence relationships with others [10,13]. This research has adopted the notion of “spirituality” from an inclusive perspective as a collective term for forms of spiritual identity (religious or non-religious in nature).

### 1.1. Spiritual Diversity in Australia

Australia is an immigration country composed of multiple races, cultures, and religions. According to the Australian Bureau of Statistics [ABS] [14], approximately 52.2 percent of Australian residents are religious, and with the increase in the number of immigrants with various religious backgrounds, religious affiliation in Australia is becoming more diverse. Data from the 2021 Census shows 43.9% of Australian residents reported Christianity as their religion, 3.2% reported Islam, 2.7% reported Hinduism, and 2.4% reported Buddhism [14]. Even though 38.9 percent of Australian residents had no defined religious identity, some considered themselves spiritual. These data demonstrate the high relevance of religion and spirituality to Australian residents and their diversity in the Australian context, proving impetus for researchers to explore the spirituality of cultural communities and the mechanisms of their role in mental health recovery. 

There has been literature on spirituality and mental health related to cultural groups. However, there has been no research to date on the spiritual perspectives of mental health service users from the Chinese community in Australia, which resulted in limited existing knowledge of this area. China is the fourth largest country of birth in Australia, accounting for 5.5% of the entire population, and Mandarin is the most spoken language in Australia other than English [15]. The authors suggest that eliciting broad information from Chinese participants from community mental health services about their spiritual identities is necessary to inform understanding and knowledge in practice for service users from this cultural group to work on their spiritual identities, underpinning the promotion of service equity. Thus, in this study, we set out to explore the following question: What are the views and perspectives of Chinese service users with mental health challenges on the role of spirituality in their personal recovery?

### 1.2. Mental Health Community Support Services (MHCSS)

Mental health services in Australia are categorized into clinical and non-clinical classifications [16]. This research was conducted in a non-clinical mental health setting. Non-clinical mental health services, also known as MHCSS [17], are based on the concept of recovery and are constantly changing, adapting, and refining in response to service users’ aspirations [16]. MHCSS provides support services to people experiencing mental health challenges in a defined geographical area throughout their recovery process [16]. 

## 2. Materials and Methods

### 2.1. Design

We adopted an exploratory research design for a qualitative analysis of recruited service users’ narratives and experiences to address the research question. 

### 2.2. Participants

The participants were recruited from the Psychosocial Support Service (PSS) program, which is a branch service of a national MHCSS organization. PSS delivers recovery support for individuals facing mental health challenges. The participants for this study were recruited through non-probability purposive sampling. Eligible participants (1) were aged 18 years or above; (2) were from the Chinese community with a spiritual identity who had accessed or were accessing personal recovery services (non-clinical); (3) could communicate in Mandarin or English. 

The research participants were recruited through email and advertisements on the PSS communication platform—methods that were facilitated by PSS practitioners who were working with service users from the Chinese community. The email recruitment significantly contributed to the recruitment efficiency, as it allowed a prepared invitation letter in plain language to share and clarify the topic and purpose of the research, criteria for participants, such as the age range and having a spiritual identity or perspective, researchers’ contact details, and other necessary information. Email recruitment also avoids the pressure of needing to respond in a timely manner or in a face-to-face scenario [18]. However, the recruitment efficiency might be affected by the degree of interest in the research among practitioners [19].

### 2.3. Data Collection

The qualitative semi-structured interviews were employed in either face-to-face or zoom online sessions with each participant to collect data. Interviews varied in duration from 60 to 120 min, depending on the conversations, and all interviews were conducted within 7 workdays. The semi-structured interviews followed the prepared interview guide, which included five general demographic questions and five open-ended questions formulated from reviewing published works: (1) “What is your understanding of spirituality personally?”; (2) “if available, could you share any stories or experiences that you have integrated spirituality into your life?”; (3) “have you had any experiences with formal religions or religious groups/organizations?”; (4) “what are your views about bringing your spiritual beliefs into your mental health recovery?”; and (5) “would you like to share anything else with us regarding spirituality-related concerns?” to elicit the Chinese service users’ perceptions of spirituality. We extended or paraphrased some of the interview questions based on the interview guide, according to the interview circumstances at the time. Following the preferences of the participants, all interviews were performed in Mandarin; three of the interviews were recorded by audio recording, and one interview was recorded by taking notes.

### 2.4. Data Analysis 

The template analysis, which is a particular approach of thematic analysis (TA), was employed in this study [20,21]. The process of template analysis is highly flexible, allowing the adoption of a priori themes to assist the researchers in entering the analysis and iterative coding of the templates to enable the exploration of the richest aspects of the data [20]. 

The template analysis was carried out in four main steps. First, the first-named researcher transcribed and translated the audio data and performed secondary manual proofreading of the accuracy of the transcripts. Second, after reading and familiarizing themselves with the entire data repeatedly, the researchers performed preliminary coding of the data [20]. These codes included a priori themes and codes, such as “meaning-making”, “coping”, and “spiritual struggles”, which were formulated from previous systematic review literature in the field [3]. Some codes were also included that might be useful for further analysis, such as “connecting with deceased loved ones” and “preventing harmful behaviors and thoughts”, as indicated in the text. Third, the research team defined and grouped the primary codes through discussions and created the preliminary template based on two of the interview data. Fourth, the authors used the preliminary template to analyze the data from the other two interviews and iteratively modified, removed, and inserted data into the themes or codes as needed [20].

### 2.5. Ethical Considerations

This research has approval from the Monash University Human Research Ethics Committee, and the researchers also obtained a permission letter from the PSS program where the participants were recruited. All participants received explicit confirmation of the research purpose, confidentiality procedures, and the right to withdraw before the commencement of the interviews [17,18]. They were informed that withdrawal from the research would not have an impact on their receiving current or future services [22]. In addition to this, interviews were conducted in settings where the participants felt comfortable (participants’ choices were prioritized).

### 2.6. Rigor

The trustworthiness of the current research was ensured through the below ways. First, an interview guide was designed prior to the interviews to ensure the reliability of data collection. Second, the first-named researcher responsible for the data transcription and translation shares a common linguistic and cultural background with the research participants, which enhanced the trustworthiness of the data. Third, the responses received from peers during several group debriefing sessions on the study’s design, methodology (especially the data transcription process), and ethics contributed to the credibility of the study [23].

## 3. Results

### 3.1. Participants

The sample size for this study was four, with participants in the age range of 24–70 years old. Two of the four participants were male, and two were female. One participant received a high school certificate, two of the four participants received a college diploma, and one received a bachelor’s degree. Three of the four participants identified themselves as Christian. Of these, two considered themselves to be devoted believers in Christianity; one considered himself to have chosen Christianity but is still exploring a deeper level of spirituality. One of the four participants identified herself as spiritual but non-religious; however, her explanation of her spirituality is presented later in detail.

### 3.2. Themes from the Interview Data

The findings of a template analysis of transcripts were analyzed and reported under the following four main themes: (1) understanding of spirituality, (2) stories/experiences about integrating spirituality into life, (3) experiences with formal religions/religious groups, and (4) views about bringing spiritual beliefs into mental health recovery. Each theme has further generalized subthemes. Summarized major findings are outlined in Table 1.

#### 3.2.1. Understanding of Spirituality

The first theme includes two second-level themes on the participants’ understanding of spirituality. Participants articulated their understanding of spirituality—an understanding that somehow integrated their self-defined spiritual identity. 

*Concept of spirituality.* All participants emphasized that they regarded spirituality as a positive presence. Some described spirituality as a felt and dynamic presence, while some tended to conceptualize it as religion and related categories. 

Participant 1: *“[Spirituality] gives people a kind of direction, a kind of guide to life, … About spirituality, I’m actually still in the process of exploring it.”*

Participant 2: *“I’m a Christian…, but spirituality, I only have a simple understanding, which is that it’s a connection between the soul and the God we believe in.”*

Participant 3: *“[Spirituality] is God, miracles, and my Christian faith.”*

Participant 4: *“For me, it’s probably the connection with my parents who have passed away…” [spirituality] is a feeling of support.”*

*Self-defined spiritual identity*. Participants defined their spiritual identity in relation to their subjective feelings and personal values.

Participant 1: “*I don’t exactly think of Christianity as unique, because often it’s just a [choice].”*

Participant 4: *“[Deceased parents]’s spirits have always been with me even though they were away, and that is my spirituality.”*

#### 3.2.2. Stories/Experiences about Integrating Spirituality into Life

For the theme of stories and experiences of integrating spirituality into lives, two highly focused second-level themes were generated from the data: reasons for becoming a spiritual believer and spiritual activities.

*Becoming a spiritual believer*. Participants shared stories about their experiences of becoming spiritual believers. The reasons stated by the participants that motivated them to become spiritual believers were linked to some aspect of their lives, such as being influenced by a positive role model from their surroundings, opting to develop a spiritual identity to find balance and stability in their lives, or mentally, when suffering from uncertainty or mental stress. It is noteworthy that, in the participants’ narratives, spiritual perspectives are dynamic as life circumstances and time change. 

Participant 1: “*I came to Christianity because I knew an aunt who is a Christian and I feel that she has a different quality of character from ordinary people, that is, although she is not very rich, she is willing to give and help others…”*

Participant 3: *“I am a third-generation Christian, I was not very religious, and did not have a deep sense of my Christianity…my deeper identification with the Christian identity emerged when I came to Australia and encountered extreme difficulties.”*

Participant 4: *“My father was a big influence on me when I was growing up… I think if my marriage was happy and easy, maybe there would be less of this kind of thinking, but then I feel aggrieved that only my parents are the most loved.”*

*Spiritual activities*. The participants, whether religious or non-religious believers in spirituality, all have their preferred spiritual activities. Under the second-level theme of spiritual activities, a broad range of individual-based, family-based, and group-based activities are covered in the third-level themes. The individual-based spiritual activities reported are prayer, rituals, meditation, listening to classical music, and engaging with nature. 

Participant 1: *“Meditation can give me a place to put a lot of aggression…I feel that if there is an activity that can make people feel at ease, that is, meditation…I might play some more positive music when I meditate, like lighter music, piano music…”*

Participant 4: *“I have a yard [with plants], no matter how big or small, and it’s nice to have a place to bask in the sun.”*

Regarding the narratives of spiritual activities, the response from a participant indicated that rituals are at times culturally distinct or/and performed by highly individualized expression.

Participant 4: *“The Chinese have a habit of making offerings at the Dragon Boat Festival, the Mid-Autumn Festival, and sometimes I go to a shop over there [a place named Box hill] that sells the Four Treasures of the Writing House, which has incense and ritual items.”*

Of these four participants, two reported that they have been attending church and online learning groups on a regular basis. One of them further reported that studying the Bible with a family member is also one of the regular spiritual activities that she enjoys. 

Participant 2: *“There are definitely prayers, and then there are activities on Wednesdays and Sundays, We discuss something that’s related to the Bible in this group.”*

Participant 3: *“Attend a one-hour online group prayer session every morning…my son and I often study/read the Bible together at home.”*

#### 3.2.3. Experiences with Formal Religions/Religious Groups

*Positive experiences*. All participants reported they had experiences of engagement with or participation in formal religious groups. Three reported that their engagement in formal religions had brought them positive experiences. These experiences included observing or personally experiencing religious groups that provide people with tangible support, such as financial assistance and interpersonal support and emotional support, such as a sense of belonging, encouragement, and trust.

Participant 2: *“They may be helping you with money, but when we first come to Australia, we are lonely or lack communication…because there is a community, and this community must be the one that leads you to the right place.”*

Participant 2: *“Then there’s my own experience of having had some problems…and then having some legal problems…the church members and priests were really keen to help you, and they would be willing to prove that you’re a good person.”*


Participant 4: *“…when I told him I was sick, he [teacher with religious belief] would pray for me…there is a good feeling towards this religion [for me].”*

One participant reflected that his experience of participating in a religious group brought him positive energy. He feels that religious groups bring people in the community together and transmit a belief of helping each other.

Participant 2: *“It means that a person influences us first, then that person gets help, gets saved, and then he helps other people out of gratitude. So having a community like this is a great strength and a great help to those who need it… I think this is help from people or God…there is something like this that brings us together to accomplish something that would otherwise be impossible.”*

*Negative experiences*. However, one participant reported having negative experiences when attending church events. These negative experiences included having conflicts of opinion with others in the religious community and feeling excluded from the religious community.

Participant 1: *“I had a bit of a problem with my previous church, with one of the elders in the church, because maybe I asked some inappropriate questions… It was a Chinese church, and some of the elders had some preconceptions about me…, so I didn’t want to go… There were some people in religion who took it more personally, and treated me in a different way rather than arguing with me.”*


#### 3.2.4. Views about Bringing Spiritual Beliefs into Mental Health Recovery

Drawing on the participants’ responses, the research team categorized four second-level themes under the theme of views on bringing spirituality into personal recovery, including positive impact, the prevention of behaviors or thoughts that harm oneself and others, spiritual struggles, and the provision of services. 

*Positive roles of spirituality in the recovery*. All participants believed that their spiritual identity has a wide range of positive influences in their recovery journey. These include coping, bringing inner strength, and promoting positive self-transformation and life choices. Of these, three participants specifically spoke of their spiritual identity contributing to their coping with the negative aspects of their mental and emotional health and/or the challenges in their relationships.

Participant 3: *“Sometimes I recall some injustice done to me by some people, some aggravation… Just being trapped inside, but if there is some positive music and then reading some positive things, I think it helps me.” (Participant 1)“My connection with God and my religious beliefs have made me softer and humbler, I have changed myself a lot, I am not so angry, and then my relationships have changed a lot too.”*

Participant 4: *“You have one great pleasure, a spiritual strength, and you can get through any bad days you have, and you don’t feel alone…My parents said that…with or without marriage, you feel bored and lonely, but you have a spiritual anchor that will always sustain you.”*

In addition, in relation to making positive life choices and personal changes during recovery, three participants indicated that: 

Participant 1: *“I think for the most part it is a good thing if one is wise enough to understand faith, to have a positive change in your life, a positive attitude.”*

Participant 2: *“I think there are some good things that I can do or not do that are not in my duty, now I’m not saying that I will do everything, but maybe once or twice I will get it done.”*

Participant 3: *“I think it [spirituality] is very important for my mental health, through prayer, with the help of God, I have a better idea of what I wanted to do…I have had a lot of epiphanies in my connection with God and I have realized that I need to be genuine and be true to myself…”*

*Preventing harmful behaviors and thoughts*. We categorized this remarkable finding from the interview data as another second-level theme. Two of the four participants explicitly reported that their spiritual identity had prevented them from harmful behaviors or thoughts toward themselves or others.

Participant 2: *“I once lived in a rented house, and because of some conflict with my parents, I set fire to a mattress, … I used to fix the people who made me angry. …[spirituality] helped me step by step, one through the church, and the other was meditation where you can feel it helping you.”*

Participant 3: *“I experienced domestic violence and my husband trampling on my dignity …and with the language barrier… I wanted to commit suicide… at my most desperate moment, I asked God if he could help me since I was a Christian. I have since been guided by God to meet many people who have appeared one after another to help me at this time in my life.”*

*Spiritual struggles*. Two participants disclosed the spiritual struggles their spiritual identity brought for them. One reported conflicting spiritual views with a family member, and the other one reported feeling discriminated against by the religious community because of a unique understanding of spirituality. They also reported negative mental emotions when their spiritual identity was not accepted or respected.

Participant 4: *“Sometimes it wasn’t a good conversation with him. He said he didn’t communicate with his parents when he was a kid …He said he didn’t feel like his parents had a big influence on him while growing up, ‘they’ just fed us and made us read, and there was no communication… I get upset, we both don’t trust each other, so he doesn’t understand what you’re going through…In fact, our two spiritual worlds are completely different.”*

Participant 1: *“As a Christian, if I keep an open mind about spirituality, it is as if I have sinned…I would get emotionally depressed sometimes because of this.”*

*Service Provision*. The fourth second-level theme is about service provision. The responses indicated that when mental health is in the recovery phase with the support of the mental health service, they have more complex and broader needs regarding their spiritual identity, such as looking for cultural understanding, wanting more space for personal discourse with workers, and expecting to be understood, empathized with, and stimulated on a spiritual level. The results showed four service users from the same cultural community held different expectations of services based on the diversity of their spiritual identities and personal values. Two of the four participants referred to talking about spirituality and spiritual-based group activities when sharing their perceptions of inclusive spirituality in mental health services. 

Participant 4: *“I would like someone to provide us with spiritual support on a one-to-one basis, rather than a group style. It makes me feel a little more comfortable. Someone who comes from a similar cultural background may understand me better.”*

Participant 4: *“I like that the facilitator of the group activities assigns me the tasks that need to be done and then lets me enjoy it at my own pace without the pressure of having to socialize or interact.”*

Participant 2: *“I don’t think it’s possible to satisfy all people’s needs for faith, those who have faith will go themselves. But I think services could be more likely to invite some people who can, for example, I know that Wellways [mental health organization] has a lot of events, where they can go and organize a sharing session, which might not be helpful to everyone, but if one or two people are willing to join, it would be helpful to them.”*

## 4. Discussion

### 4.1. Lessons from Participants 

The interviewed four participants described their understandings of spirituality and spiritual identity, recognizing the positive impact that spirituality presented on their lives and mental health. However, they had different views and perceptions of the scope and roles of spirituality. Coping, self-regulation, and social support mechanisms were frequently noted in relation to promoting mental health recovery. On the other hand, two participants highlighted that the differences in spiritual views with family members or others in the religious community could be a source of spiritual struggles and sometimes detrimentally affect their mental health. Moreover, these mechanisms of action are culturally embedded in some measure when they positively or negatively impact recovery. A similar result was shown by Weber and Pargament (2014) [5], that religion/spirituality may damage mental health through misunderstandings and conflicting views. 

The findings of this study suggest that there may be considerable room for development in the current practice in terms of discussions with service users’ spiritual identity. All four interviewed participants expressed a willingness to discuss their spiritual identity within the services. However, cultural and language differences, as well as differences in individual expectations about forms of exploration, can be potential barriers. 

### 4.2. Spirituality in Recovery: Commonality and Individuality

Three of the four participants had a spiritual identity as Christian. This spiritual identity is consistent with mainstream Western religions and values [14]. This may reflect the trend that Australian immigrants living in a culture that emphasizes difference and diversity are also contributing to the creation of new narratives of national identity [24]. However, as Lepherd (2015) noted [25], spirituality is understood and expressed differently by individuals and may vary within a religious framework. For example, two participants viewed spirituality as a synonym for religion and believed Christianity was their full affiliation. However, one participant indicated that Christianity is their personal choice, and they are open to exploring spirituality broadly. Individualized exploration within a religious framework may pose challenges to integrating into a homogeneous religious community or even create a sense of exclusion for the ‘explorer’.

Spiritual identity may also vary, even among people of the same ethnicity and culture [25]. In conceptualizing spirituality, one participant affirmed the eternal and spiritual nature of the soul, explaining that they related their spiritual identities to this survival of the soul but were not religious [26]. The above discussion reflects the diverse and unique understandings of spirituality among people within a shared cultural or religious identity [26].

The four Chinese service users also reflected commonalities and individualized components when describing their spiritual experiences and the role of spirituality in recovery. First, they shared their spiritual identity and experiences in a way that was consistent with their personal reality, and their descriptions often combined their cognitions, perceptions, and personalized needs [12]. Furthermore, these four participants all reported that spirituality or spiritual activities brought them feelings of peace. This reflects the positive effect that spirituality may have on the mental health of Chinese cultural groups through coping [3,12,27] and self-regulatory mechanisms [25]. This is in line with the results of the study by Ho et al. [25] that peace and tranquillity are specific to the Asian groups’ perception of spirituality in their recovery. However, due to the diversity of life experiences and mental health challenges, three of the four participants’ understanding of the role of spirituality in recovery also contained other individualized interpretations, with social support and personal growth being the most emphasized.

Notably, the four Chinese service users of the PSS program rarely reported that spirituality promotes recovery through mechanisms such as bringing meaning to life, which is inconsistent with the findings in much of the literature that meaning-making is a key theme in the spiritual experiences of people with mental health challenges [3,27]. Factors such as ethnicity, country of origin, culture, and values, may have contributed to the variation in this result. However, the sample size of this study was limited, and the gap revealed by the finding should be further examined.

### 4.3. Cultural Embeddedness of Spiritual Identity and Practice

The findings demonstrate the important influence of cultural factors on the four Chinese service users’ understanding and views on spiritual identity and practice. First, they commonly believed that spirituality facilitated self-regulation, while spirituality brought positive improvements for their recovery by returning them to an inner state of ‘peace’ and ‘tranquillity’. This may reflect the fundamental Chinese traditional values of ‘acceptance’ and ‘letting nature take its course’. Additionally, two participants, who were older first-generation immigrants, might be more devout than younger Chinese participants to certain traditional Chinese values, which is entirely consistent with the findings of Revens et al. [7] in their study focusing on Latin American immigrants. For example, traditional values based on family and blood relations are sublimated as a form of spiritual identity. Based on the above discussions, it is recommended that mental health practitioners should provide creative understanding and cultural awareness when discussing with service users their spiritual identities, perspectives, and spirituality in the wider context [28].

However, cultural embeddedness refers not only to the influence that the original culture and values have on the four participants but also to how Australian culture shapes their views. Even though the four service users have cultural identities from their original country, “they join the nation they too join a national, inherited culture” [23] p2168.The results from this research show that this phenomenon is reflected more in the second generation of immigrants, meaning that they are more open-minded and democratic towards spirituality, which reflects Australian values. They are more willing to explore and debate when they encounter religious issues. This illustrates that the cultural embeddedness of spiritual perspectives in multicultural communities may be bi-directional. Additionally, if workers are able to recognize the strengths of recovery in both directions of culture, stronger and more effective support will be provided for the service users’ recovery. 

### 4.4. Implications for Practice

This study has implications for organizations providing similar psychosocial services and also for practitioners and service users in this field. The findings of the study present the views and needs of the spirituality of four Chinese service users with a spiritual identity to service providers and practitioners and have advocacy implications that may provide evidence for further policy development or service planning by organizations. Additionally, the findings provide a practical reference for practitioners’ interactions with ethnically diverse service users with spiritual identities and are beneficial in bridging the gap in understanding and interpreting spirituality between practitioners and service users from diverse backgrounds, particularly those from Asia, and in promoting equality, respect, and inclusion in-service practice.

The catchment area for the PSS program is multicultural and, in particular, covers a high percentage of residents from Asian backgrounds. The findings of this study provide the views and perspectives of four Chinese service users on the integration of spirituality into services. The findings will contribute to facilitating ‘productive interactions’ between practitioners in the PSS or mental health services similar to their function and service users in the Chinese community.

### 4.5. Recommendations for Further Research

Further studies should explore a larger, ethically diverse range of participants’ views about how spirituality forms a basis for how they interact with services and interact with their recovery processes. While the strength of this study was the rich depth of personal stories and narratives through the modest number of Chinese participants provided, further research will benefit from a larger number of participants and a broader range of cultural backgrounds.

### 4.6. Strengths and Limitations 

The present study has several strengths. Despite the challenges imposed by a pandemic, and the limitations of study establishment, ethical considerations, and data collection within a setting that was required to change to work partially remotely, four participants in a ‘real world’ setting who are currently using services, were recruited. They provided valuable and meaningful data that will contribute to the enrichment of literature, locally and internationally. This is the first evaluation in Australia of participants from this program, and most importantly, this is the first study in Australia to provide a depth of spiritual experiences and preferences from an Asian and, in this case, Chinese perspectives and lenses. 

There are two main limitations to the study. Given the ongoing challenges of a pandemic, people had reduced availabilities and restricted time to participate in face-to-face interviews. The researchers’ ability to recruit a large number of participants was impacted. The generalisability of the findings in this study may be limited due to the modest sample size and the fact that all recruitment happened through one service in one organization.

## 5. Conclusions

This study is the first to explore the spiritual views of Chinese service users within an Australian MHCSS setting. This study emphasizes that service users’ culture, values, and personal experiences, which may associate with migration, will influence their understanding of spirituality and their perceptions of its role in relation to recovery. More importantly, the current study demonstrates its role in supporting personal recovery, particularly in coping with mental difficulties and assisting self-regulation. This study also conveys the positive attitudes of Chinese service users towards discussing spirituality in services, which may resonate with service users in the wider multicultural community.

## Figures and Tables

**Table 1 ijerph-20-02210-t001:** Final template codebook.

1. Understanding of Spirituality
1.1. Concept of spirituality
1.1.1. Life direction, guide, and support
1.1.2. A sense of security
1.1.3. Connection with God
1.1.4. Connection with deceased loved ones
1.2. Self-defined spiritual identity
1.2.1. Strict Christian
1.2.2. Christian but still exploring
1.2.3. Connection with deceased loved ones (Spiritual but not Religious)
2. Stories/experiences about integrating spirituality into life
2.1. Reasons for becoming a spiritual believer
2.1.1. The attraction of the spiritual traits of spiritual believers
2.1.2. Helplessness or turning point in life
2.1.3. Born in a traditional Christian family
2.1.4. The impact of the values of the deceased loved ones
2.2. Spiritual activities
2.2.1. Individual activities
2.2.1.1. Prayer
2.2.1.2. Rituals
2.2.1.3. Meditation
2.2.1.4. Classic/piano music
2.2.1.5. Engaging with nature
2.2.2. Family activities
2.2.2.1. Learning the Bible with a family member
2.2.3. Group activities
2.2.3.1. Group online learning
2.2.3.2. Attending regular church
3. Experiences with formal religions/religious groups
3.1. Positive experiences
3.1.1. Intangible support
3.1.1.1. Financial support
3.1.1.2. Interpersonal support
3.1.2. Emotional support
3.1.2.1. A sense of belonging
3.1.2.2. Encouragement
3.1.2.3. Trust
3.1.3. Positive energy
3.1.3.1. Bringing people together
3.1.3.2. Spreading the belief of helping each other
3.2. Negative experiences
3.2.1. Conflicts with others of the religious community
3.2.2. Feeling of exclusion from the religious community
4. Views about bringing spiritual beliefs into mental health recovery
4.1. Positive roles of spirituality in the recovery
4.1.1. Coping
4.1.2. Inner strength
4.1.3. Positive self-transformation
4.1.4. Positive life choice
4.2. Preventing harmful behaviors and thoughts
4.2.1. Suicide
4.2.2. Self-harm behaviors and thoughts
4.2.3. Revenge on others
4.3. Spiritual struggles
4.3.1. Spiritual conflict with family members
4.3.2. Feeling stigma and discrimination from communities
4.4. Service provision
4.4.1. Talk about it
4.4.1.1. One-to-one support from worker
4.4.1.2. Workers from a similar cultural background
4.4.1.3. Obstacles created by not speaking English
4.4.2. Spiritual-based activities/groups
4.4.2.1. Religious groups
4.4.2.2. Activities that are more creative and less sociable

## Data Availability

Please contact the corresponding author.

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
