# Peer review of "Spiritual Diversity in Personal Recovery from Mental Health Challenges: A Qualitative Study from Chinese-Australian Service Users’ Perspectives"

_ijerph, 2023, doi:10.3390/ijerph20032210_

Round 1

Reviewer 1 Report

Very interesting article, but needs further study and validation of other questionnaires and questions. Review the references.

Author Response

Thank you for the interest in the research; we agree that more study in this area is warranted. The current research was limited in scope due to being undertaken within a Masters research project. We look forward to conducting research on this topic with more questions in future. To flesh out this paper we have now added considerably to the literature in the Introduction.

References have been reviewed. 

Reviewer 2 Report

The authors offer an interesting qualitative research on the diverse understanding of spirituality of Chinese users of mental health service in Australia. In general, the writing is clear and sounds.

Few comments are listed as follows:

The introduction section is quite general and is not able to set the scene for readers. For example, the first paragraph mention about the challenges to mental health of spirituality without further explanation; the second paragraph mention about spirituality could be religious or non-religious in nature while most of the section 1.1 is only about the demographics of religious belief of Australian. Besides, the research gaps should also be stated in a more explicitly manner.

As suggested, this work is stated as an exploratory research and could be the first research of its kind in this population. It is fair to affirm the importance of spirituality and suggest to integrate spirituality to service. I still wonder the RQ2, “HOW could practitioners engage…” is sufficiently addressed.

In the discussion, as an exploratory research, I wonder if it is appropriate to make conclusive sentence “there is a high rate of Christians among service users with a spiritual identity in the Chinese community.” Also it might be a bit dangerous to be so conclusive for the generational difference in section 4.2 based on the investigation reported in section 3.

Also a minor comment for the authors’ consideration:

Since this research is about the Chinese users and one of the participant reports about the “language barrier”, I wonder the authors would also like to report the use of language in interview and the way to handle the data in the methodology.

Author Response

  1. a) The introduction has been further developed by integrating more literature to present the readers with the background and status of research on the topic. The authors have extended on the content on the challenges of spirituality for mental health and supported it with more citations. b) and c) The authors have adapted the structure and content of the 1.1 Spiritual Diversity in Australia to clearly present research gaps and the importance of the research topic in the Australian context.
  2. The research questions have been adjusted to reduce the formal research questions to one which is : “What are the views and perspectives of Chinese service users with mental health challenges on the role of spirituality in their personal recovery?” This can be viewed at the end of 1.1 Spiritual Diversity in Australia. The original question 2 has been placed in the discussion section and presented as suggestions and inspirations for future services (p14-15).
  3. a) & b) Given the typology of the research and the small sample, the authors revised all the generalized conclusions to present them in a more objective form.
  4. In the methodology section authors added a clarification that the language used for the interviews is Mandarin and it as also confirmed how the data is transcribed and translated in the “data analysis” part. Additionally, the authors added a paragraph on rigor to show how to ensure the accuracy of the translation re data. (p5).

Reviewer 3 Report

This manuscript has many strengths, including the use of qualitative methods for this specific study and the worthwhile topic. I will note specific strengths in my comments below.

At the same time, there are some ways that the manuscript can be strengthened in each of the main sections.

Review of Literature: The review of literature is solid on spiritual diversity in Australia, providing a nice local context for the method, especially since many writers feel that qualitative research should be interpreted *contextually* (e.g., that a particular view of social reality is linked strongly to the contexts where it is lived—which is one of the main rationales for doing qualitative inquiry), rather than with “generalizing” claims (see notes on discussion). There is a little detail, but not much development on mental health and support services, with a clear distinction between clinical and non-clinical mental health.

Where the review of literature needs strengthening, however, is the main argument that drives to the RQs for this study, specifically, what is the connection between spirituality and mental health (in previous literature), what is the link between either of these two and cultural adjustment (since you are focusing on transnational individuals), and (based on the specific RQ questions for this study) why those should be studied specifically with a group of *Chinese* transnationals in Australia, say as opposed to immigrants in general. That is, in its current form, the review of literature does not provide a solid support for these research questions.

The second RQ (“How could practitioners engage…”) seems more like a question of *interpretation and application* of the findings rather than a question you will answer directly with what the data say. I would focus RQs on what you can “know” from the data.

Method: The use of qualitative methods to gain nuanced perceptions of participants on a topic for which there is little academic research is well-warranted in previous literature, and this sort of data can provide rich insights, even if it is not just “exploratory” (as the article states). If it is exploratory, I would expect the discussion to say where these “exploratory” findings can go next in terms of further validation. Purposive sampling is appropriate, following great qualitative “giants” as Lincoln and Guba or other standard qual textbooks (the reference list lists a social work text on methods, which is good—but you could support this with broader literature justifying the method). (Note: I didn’t see “source 13” cited in the text, unless I missed it.). It seems that the research was purposive only in that there were criteria (what some have called “criterion” sampling). But why choose just Chinese (see note on review of lit—if there is a stronger rationale there for Chinese participants, this issue is resolved)? Beyond the criteria of inclusion, was the sample a “convenience” sample, or did you ,for example, seek “maximum variation” among the Chinese participants (Lincoln & Guba, Naturalistic Inquiry, 1985, and other texts).

Just a format note, but I would give “participants” the same level heading as other aspects of the method. I’m not sure what you mean by “confirmation with potential participants.” One major concern is the small amount of data for the study—four interviews. They are longer interviews (60-120 minutes, though we don’t have an average time), but this is fewer than most scholars who write qual methods texts suggest. I would recommend more participants for greater trustworthiness of the data. If that is not a possibility, address the “elephant in the room” by admitting the small sample with source justification for why this is an adequate sample. Move note about recruitment efficiency up to the discussion of recruitment.  Important for the discussion, mention what you were looking for in the invitation email. For example, did you explicitly seek Chinese immigrants who had had experience with the mental healthcare (nonclinical) system in the invitations to participate? How you recruited participants impacts the sorts of claims you can make in the interpretation of the findings.

The “semi-structured” interview method (quite appropriate) inherently assumes that you will ask the questions different ways, in different orders, or with different follow-up questions with each participant, so this information serves more as a *definition of* the semi-structured approach than a contrast to it (“however”). It would be good to include more detail on the actual questions asked—a paraphrase of the main line of questioning.

The data analysis seems reasonable. In the abstract, you mention “King’s Template Analysis,” though that phrase does not come up in the article. What is King’s Template Analysis? It’s great that you mention that you bring some “codes” into the analysis a priori (don’t forget the “a” in “a priori”); what is the source of these codes? Ethics: Explain *how* you explained and obtained consent “in advance” (e.g., did you send the participants the consent form and have them give consent at the beginning of the interview?).

RESULTS:

Consider moving participant demographics to “participants” section of method (though some authors do put such information in the beginning of the results). The main categories (presented nicely in tables and then in text) make sense in terms of structure and substructure. As you go along (or in the discussion section), make it clearer how these relate to the RQs asked. The findings follow the figure well. (Note in outline, generally I don’t expect to see one subheading of an item if there is not at least one other subheading at the same level, e.g., 4.3.1).

As I teach my own qualitative students, I encourage a balance between interpretation and exposition of direct quotations (some indented as block quotations, some as quotations inside of paragraphs, and some “paraphrased” by the author). No quotation should be in the text without some interpretation or mention of it as it relates to the category. E.g., if there are quotes laid side by side (e.g., pp. 5-6), then the text should tease out the nuances between these. More often, I would recommend including each exemplar as it supports a claim.

This would prevent one of the limitations I see in the exposition: Some quotations (or portions of longer quotations) don’t seem strongly linked to the categories that they are supporting (e.g., P #4’s elaboration on the dress her mother made for her, lines 225-228). In most cases, the delineation of points and exemplars work together. But there are places where the exemplars seem to run ideas together. One such place is category 3.6 (p. 8).

Ironically, this gets at one of my greatest recommendation for the methods. There is a lot of material here that talks about what spirituality is and elements of religious or spiritual practice that are unrelated to the RQs—about the use of religion or spirituality in terms of *mental health*. Point 3.6 gets the reader closer to the research question, but is one of the sections most characterized by a stream of unexplained/uninterpreted exemplars. These exemplars seem to touch on a variety of themes. So, do more to explain how these relate to the theme. I recommend at the beginning of this section (and each section) a clear statement about the definition of the “main-level theme”—here, “bringing spirituality into personal recovery.” Make sure, by the end of the “results” on these sections, that we have a good sense of how the findings relate, specifically, to mental health recovery.

DISCUSSION: The section has good development, with clear relation to previous literature. Some qual researchers resist making firm statements of cause-and-effect with qualitative data (which tends to work from an “interpretive,” rather than a “social scientific” framework (e.g., “the findings suggest that spirituality has positive impact on the recovery of Chinese service users”…, lines 354-355). In addition, I would avoid generalizing statements. Four participants, engaged in semi-structured interviews that strongly contextualize the findings on the nature of the interaction between the participants and the researcher, are hardly enough participants to make the generalizing claim that the study findings show that “there is a high rate of Christians among service users with a spiritual identity in the Chinese community” (lines 369-370). In fact, if there is such a large number of religious Chinese in the Australian context, one would wonder why you have only four participants.

The discussion top p. 10 and following is good, especially as it relates to “cultural embeddedness” in the Australian culture as this relates to the Chinese cultural values that the participants bring with them (indeed, this unique cultural blend might be the sort of rationale you need for a focus on Chinese residents in Australia in the rev of lit).

You note that “Chinese service users also reflected commonalities and individualized components when describing their spiritual experiences” (388-389). I like this nuance; however, it would help, noting their common base in spiritual or religious identity, to know what the invitation asked for. If, for example, you specifically invited participants who considered themselves as spiritual or religious, this would explain why the participants have this commonality.

I really like the practical implications at the end. This would be a great place for even more specific recommendations related to RQ 2 (“How could practitioners engage with the spirituality of Chinese service users”). As noted above, I don’t see this as a research question (unless you specifically *asked* participants about their recommendations for practitioners and explore this as a category set). But it is a great place to explore concrete, practical actions. That is consider switching RQ2 to more of an “application” question for the discussion section than a RQ to guide the study. Nice consideration of directions for future research that clearly address my note above regarding this as an “exploratory” research.

Author Response

  1. a) , b) and c) The authors restructured and expended the literature review to illustrate : (1) the relationship between mental health ( recovery ) and spirituality; (2) the different cultural groups may have different perceptions of spirituality and of the mechanisms by which it works in their recovery; and (3) the proportion of the Australia-Chinese- population in Australia and the significant lack of attention to this community in this field.
  2. The authors adjusted the research questions-keeping the first RQ and leaving the original second RQ ( “ how could practitioners engage…) as an interpretation and application in the discussion section which is more reasonable. The data collected only provided direct content for the first RQ.
  3. a) the discussion now states where these “exploratory” findings can go next. b) thank you for recognizing the use of a methods text; we have expanded on how theory is used here in the explanation of the approach taken to the research. c) Source 13 refers to a government website that provided evidence to distinguish between clinical and non-clinic mental service in Australia. This has been amended in the manuscript. d) The authors explanation as to why the Chinese group was targeted for the study are presented in the developed literature review. f) This comment again refers to the literature that suggested by reviewer (Lincoln & Guba, Naturalistic Inquiry, 1985).

  4. Format has been revised in the revised manuscript as suggested. “confirmation with potential participants” was removed, however authors re-edited this section to present a clearer picture for the recruitment process. 

    No additional data collection is possible for the current study, due to scope of data collection within a Masters qualification. The small sample size is now more clearly acknowledged in the manuscript.

    For the recruitment efficiency, authors discussed this in more detail. The content of the recruitment emails has been clarified (which contributed significantly to recruitment efficiency).
  5. In the 2.3 Data collection section, the authors have provided a description of how questions had been asked when conducing interviews (including the need to explain the main line questions or additional detailed questions as needed based on the interview situation), example question also included in the revised manuscript.
  6. The authors have adapted “King’s template analysis” in the abstract to “Template analysis considering that this research intended to emphasize an analysis method rather than the developer of the method.

    The authors have emphasized the source of a prior code (from a published systematic review) in the revised version.

    The authors have also clarified in the ethical considerations about when and how informed consent was obtained.
  7. The results on the demographic data have been kept in the Results section; the senior author on this paper has been trained that a method section describes the approach taken and all results gained – including demographic ones – are Results of the study. Thank you to the reviewer for the suggestion, and also for the choice to rest with the authors.

    The issues with the headings and subheadings in the table has been addressed, as suggested. 
  8. The authors have made a number of changes and adjustments to this section and added more subheadings (secondary themes) to make the results of the study more organized and clearer (please refer to the result section).
  9. The authors have simplified some of the quotations and removed irrelevant ones. The authors also adjusted the wording or content to avoid putting ideas and examples together (please refer to the result section).
  10. The authors have adapted the narratives of some of themes and added some explanations to enhance the relevance of the examples to the research topic and RS.

  11. The authors have modified the following changes to the discussion section: (1) streamline findings that do not have high relevance to the research topic (although it may be interesting or valuable); (2) avoid generalizing statements and instead focus on the participant perspective or conjectures generated based on the results/data- authors removed the statements such as “there is a high rate of Christians among service users with a spiritual identity in the Chinese community” which should not be generalized with a small number of sample.
  12. Thank you to the reviewer for the acknowledgement of the discussion section on cultural embeddedness.

    Authors has abridged and improved this part in the revised manuscript. – for example, unfounded comparisons of religious identities were removed, focusing instead on the diversity of spirituality and the difficulties that the non-mainstream nature of their religious identities may present for mental health/ mental health recovery.  
  13. The authors appreciate the reviewer suggestions related to the readjustment of the research questions. The second RQ, we fully agree is more appropriately explored in the section on practical implications rather than as a separated question (especially as this also has limited relevance to the data collected).  Therefore, as suggested, RQ2 has been transferred to the application question in the discussion section.

Round 2

Reviewer 2 Report

The authors has addressed all of my concerns. I believe this is reaching the standard of the journal.

Author Response

We are please with the advice that we have addressed all of concerns and the paper is reaching the standard of the journal.

Reviewer 3 Report

I feel that, in most regards, the revision meets my primary concerns with the first version. The writing is much improved, with just a few things to note, which I list below.

There is a clear section of the rev of literature that now gives a rationale for why to study Chinese immigrants in Australia.  There is more of an attempt not to generalize the findings from four in-depth interviews to all Chinese in Australia, though this type of thinking is still in the article. For example, “A high percentage of four participants” (line 868) sounds like a sort of “naïve quantification” (Berelson, 1952) that could leave less-informed readers with  notion of generalization even though you have not made a generalizing claim. Rather, I would just say “Three of the four participants had a spiritual identity…”

There is a seeming contradiction on p. 10, where, at one point you note that the findings suggest that spirituality may have a positive mental health impact on Chinese immigrant groups, but then not that the Chinese in the PPS program “rarely reported themes related to spirituality” (898-899). The findings section shows four subheadings of concepts regarding the (positive) role of spirituality in mental health recovery, with no qualification that there were limited links or themes related to spirituality. Even if there were not many comments regarding spirituality and meaning, this could be a result of the interview structure. As is, we still have little detail on what you asked in the interview. You provide one of two of the ideas from interviews that ranged from 60-120 minutes.

My main recommendations at this point would be to (a) give more detail still about what sort of questions were in the interviews (maybe attaching the interview protocol as an attachment) as well as what was included in the invitation (for example, did the invitation to participate specifically mention that the researchers were looking for participants with a spiritual perspective?); (b) clarify a possible contradiction in the discussion with the findings; (c) work more to present the findings in the discussion as a representation of the perspectives of these four individuals (not the “majority” of people interviewed). The strength of the findings is, as you note, showing the divergence of experience, even among these four individuals.

Here are some minor edits, listed by line number:

·       34-35:  views…that hindersà that hinder

·       36: bidirectional nature: I’m not sure what you mean, here.

·       43: “…identity and the social support that derives from their connection to the religious community is one of…” There are two nouns—identity and social support—so either change “is one of” to “are some of” OR separate the “and…community” clause as a “parenthetical clause” (perhaps “as well as the…community,” separating it on either side with a comma. If you do this, then “the social support” does not contribute to the number of the verb (the verb remains singular to go with “identity.” To clarify:

o   “Justice, as well as equality, is an important value” but

o   “Justice and equality are important values”

·       65-66: “cross-cultural control study”—I’m not sure what you mean

·       68: “and interpreting”à “and allowing the to interpret…”

·       70: “faith communities” à “people in faith communities” [people have “perceptions” rather than communities or groups having them)]

·       76: “the statistical limitation: I’m not sure what you mean here.

·       84: “sprituality” à possibly “the notion of spirituality”?

·       176 ff: “the participant for this study was”—do you want plural here?; “Eligible participants were aged 18 years or above” (delete “of”); Chinese (spelling); à”had accessed or were accessing” (keeping plural of the sentence subject). “Can” à “could communicate” (keeping past tense of earlier verbs, even though we assume that the participants can still speak Mandarin or English).

·       182: possibly comma between platform & methods?

·       183-184: “contributes…allowed”à keep verb tense consistent (probably past tense in the methods section).

·       187: “…in a timely manner or face-to-face”—the grammar seems incomplete

·       187-188: I’m not sure what you mean by the last sentence.

·       192: “completed within 7 workdays” –a bit confusing

·       210-211: Probably: “Second, after reading the entire data repeatedly to familiarize themselves with the data…”  This sounds like you might be referring to Braun & Clarke’s steps for data analysis, which is very appropriate. If so, be sure to include in the references.

·       323: à “two of the interviews”. Probably, “Fourth, the authors used the preliminary template to analyze the data from the other two interviews”

·       328: “…and also obtained…” à “and researchers also obtained…”

·       331: “…withdrawal from the research will not…” à “would not” (past tense, to agree with the rest of the discussion here).  I would add another phrase (for clarity): “They were informed that withdrawal from the research would not…”

·       340-341: “…the accuracy… received response from professional(s?) in the field…” I’m not sure what you mean here. If you had other professionals validate the trustworthiness of the findings, explain how you did this. Perhaps cite Lincoln and Guba (1985)—if you have access to this source (Naturalistic Inquiry) or to another qual methods text that supports “peer validation”

·       357: “Views” à “views” (to be consistent in capitalization with other items)

·       358: à “are outlined”

·       500: à “when I came to Australia.”

·       502: “…my” à “My…” [APA allows you to delete initial or terminal ellipsis periods]

·       763 ff: “him/he”:  The pronouns from the exemplar need context in the description that precedes them—who does “he” or “him” refer to?

·       788: remove comma before “actually”

·       880: “vary under the same ethnicity” à “vary even among people who have the same ethnicity”
